# Predictors for Early and Late Death in Adult Patients with COVID-19: A Cohort Study

**DOI:** 10.3390/ijerph19063357

**Published:** 2022-03-12

**Authors:** Yung-Feng Yen, Shang-Yih Chan, Chu-Chieh Chen, Chung-Yeh Deng

**Affiliations:** 1Section of Infectious Diseases, Taipei City Hospital, Taipei 10341, Taiwan; 2Institute of Public Health, National Yang Ming Chiao Tung University, Taipei 30010, Taiwan; 3Department of Health Care Management, National Taipei University of Nursing and Health Sciences, Taipei 11219, Taiwan; dad89@tpech.gov.tw (S.-Y.C.); chuje@ntunhs.edu.tw (C.-C.C.); 4Department of Education and Research, Taipei City Hospital, Taipei 10341, Taiwan; 5Department of Psychology and Counseling, University of Taipei, Taipei 10048, Taiwan; 6Department of Internal Medicine, Taipei City Hospital, Taipei 10341, Taiwan; 7Institute of Hospital and Health Care Administration, National Yang Ming Chiao Tung University, Taipei 30010, Taiwan

**Keywords:** COVID-19, mortality, timing of death, prospective study

## Abstract

The timing of death in patients with coronavirus disease 2019 (COVID-19) varied by their comorbidities and severity of illness. However, few studies have determined predictors of mortality with respect to the timing of death in infectious patients. This cohort study aimed to identify the factors associated with early and late death in hospitalized COVID-19 patients. From 14 May to 31 July 2021, this study consecutively recruited laboratory-confirmed COVID-19 patients admitted to Taipei City Hospital. All patients with COVID-19 were followed up until death or discharge from the hospital or till 13 August 2021. Mortality in such patients was categorized as early death (death within the first two weeks of hospitalization) or late death (mortality later than two weeks after hospitalization), based on the timing of death. Multinomial logistic regression was used to determine the factors associated with early and late death among such patients. Of 831 recruited patients, the overall mean age was 59.3 years, and 12.2% died during hospitalization. Of the 101 deceased, 66 (65.3%) and 35 (34.7%) died early and late, respectively. After adjusting for demographics and comorbidities, independent predictors for early death included age ≥ 65 years (adjusted odds ratio (AOR) = 5.27; 95% confidence interval (CI): 2.88–9.65), heart failure (AOR = 10.32; 95% CI: 2.28–46.65), and end-stage renal disease (AOR = 11.97; 95% CI: 3.53–40.55). This study found that two thirds of COVID-19 deaths occurred within two weeks of hospitalization. It suggests that hospitalized patients with COVID-19 should be treated carefully and monitored closely for the progression of clinical conditions during treatment, particularly in older patients and in those with comorbidities.

## 1. Introduction

Coronavirus disease 2019 (COVID-19), caused by the severe acute respiratory syndrome coronavirus 2 (SARS-CoV-2), has led to a global pandemic since January 2020. As of 4 February 2022, 383.5 million individuals worldwide were infected with SARS-CoV-2, with a death toll of 5.7 million [1]. 

The SARS-CoV-2 infection could cause long COVID-19 syndrome, including symptoms of fatigue and dyspnea lasting for months after the acute infection [2]. Since SARS-CoV-2 is highly contagious, personal protective equipment (PPE) use is extensively recommended for the general population and healthcare workers to prevent the spread of the virus and the COVID-19 infection [3,4].

While many patients experience mild symptoms [5], SARS-CoV-2 could cause an acute life-threatening disease in immunocompromised or hospitalized patients [6]. A prospective study involving 41 hospitalized patients with COVID-19 in China showed that the clinical condition in hospitalized COVID-19 patients could deteriorate rapidly [7]. The median time from hospital admission to acute respiratory distress syndrome in those patients was only two days [7]. Another prospective study in France showed that the median time from intensive care unit (ICU) admission to death was 14 days in patients with COVID-19 [8]. Moreover, the timing of death in infectious patients varied according to comorbidities and severity of illness [8]. However, few studies have determined the predictors of mortality with respect to the timing of death in patients with COVID-19. 

Understanding the factors associated with mortality with respect to the different timings of death could provide evidence to guide the clinical management of COVID-19 patients. Therefore, we conducted this cohort study to identify the factors associated with early and late death in hospitalized COVID-19 patients in Taipei, Taiwan. 

## 2. Methods

### 2.1. Background Information 

In mid-May 2021, Taiwan experienced its first huge outbreak of SARS-CoV-2, which was particularly severe in Taipei [9]. COVID-19 in Taiwan is a reported infectious disease. Healthcare institutes must report a new diagnosis of COVID-19 to the Taiwan CDC within 24 h through an Internet-based notification system. Patients infected with SARS-CoV-2 in Taiwan must be admitted to designated COVID-19 hospitals for further treatment. 

Taipei City Hospital (TCH), the largest healthcare organization in northern Taiwan (with 4700 beds), is a designated healthcare institute that accommodates patients diagnosed with COVID-19. Patients infected with SARS-CoV-2 admitted to TCH are cared for by a designated healthcare professional team. 

### 2.2. Study Subjects

This cohort study consecutively recruited patients aged 18 years or older who had received a COVID-19 diagnosis and were admitted to TCH between 14 May and 30 July 2021. The diagnosis was confirmed using a positive real-time reverse-transcriptase polymerase chain reaction (RT-PCR) test. Patients with COVID-19 transferred to other hospitals or those aged <18 years were excluded from this study. All patients with COVID-19 were followed up until death, discharge from the hospital, or till 13 August 2021, whichever applied to the patient. This study was approved by the Institutional Review Board of Taipei City Hospital (no. TCHIRB- 10904014-E). 

### 2.3. Outcome Variables

The outcome variable of interest was the treatment outcome, which was categorized as successful treatment or mortality. Mortality was categorized as early or late death, based on the timing of death. Early death was defined as death within the first two weeks of hospitalization, and late death as mortality later than two weeks after hospitalization.

### 2.4. Covariates

The covariates included sociodemographic characteristics and comorbidities. The sociodemographic characteristics included age and sex. Comorbidity was determined based on patients’ medical records, including presence of cancer, heart failure, cerebrovascular disease, diabetes, and hypertension. 

### 2.5. Statistical Analyses

First, the demographic data of the participants were analyzed. Continuous data were presented as mean (standard deviation (SD)), and one-way analysis of variance (ANOVA) was used for intergroup comparisons. The proportion of each outcome was compared according to individuals’ demographics and comorbidities. Then, we used Pearson’s χ^2^ test to analyze the categorical data as appropriate.

We assessed the crude associations of factors associated with mortality by computing odds ratios (ORs) and the corresponding 95% confidence intervals (CIs). Multivariate analysis was used to identify the factors associated with mortality among COVID-19 patients. We conducted a subgroup analysis to determine the factors associated with mortality among COVID-19 patients after stratifying the patients according to sex. Moreover, we used multinomial logistic regression to determine the factors associated with early and late death among such patients. Adjusted ORs (AORs) with 95% CIs are reported to indicate the strength and direction of the association. All data management and analyses were performed using SAS 9.4 statistical software package (SAS Institute, Cary, NC, USA).

## 3. Results

### 3.1. Participant Selection

This cohort study included 861 COVID-19 patients admitted to the TCH between 14 May and 31 July 2021 (Figure 1). After excluding patients who were transferred to other hospitals (*n* = 25) and those aged <18 years (*n* = 5), the remaining 831 patients were included in the analysis. The overall mean (SD) age was 59.3 (16.0) years, 49.0% of the patients were men, and 12.2% died during hospitalization. Of the 101 deceased COVID-19 patients, 66 (65.3%) died early and 35 (34.7%) died late. 

### 3.2. Characteristics of Patients by Treatment Outcome

Table 1 shows the patient characteristics according to the treatment outcome. Compared to COVID-19 patients with successful treatment, those with early or late death were older and more likely to be male. In terms of comorbidities, patients who died during hospitalization, had a higher proportion of cancer, heart failure, and end-stage renal disease. During hospitalization, patients with early or late death were more likely to be admitted to the ICU and to receive intubation treatment. 

### 3.3. Factors Associated with Mortality in Patients with COVID-19

Table 2 shows the univariate and multivariate analyses of factors associated with mortality in patients with COVID-19 during hospitalization. After controlling for demographics, comorbidities, and severity of the disease, independent predictors of mortality included age ≥ 65 years (AOR = 6.47; 95% CI: 3.80–11.02), heart failure (AOR = 11.67; 95% CI: 2.87–47.49), and end-stage renal disease (AOR = 18.67; 95% CI: 6.42–54.30).

### 3.4. Subgroup Analysis for the Factors Associated with Mortality in Patients with COVID-19

We conducted a subgroup analysis to determine factors associated with mortality among patients with COVID-19 after stratifying the patients according to sex. After controlling for demographics, comorbidities, and severity of COVID-19, age ≥ 65 years was found to be associated with a higher risk of mortality in male and female patients with COVID-19 (Appendix A). Moreover, heart failure and end-stage renal disease were the predictors for mortality in male patients with COVID-19.

### 3.5. Factors Associated with Early and Late Death in Patients with COVID-19

Multinomial logistic regression analysis was used to identify predictors of early and late death in COVID-19 patients. After controlling for demographics, comorbidities, and severity of COVID-19, risk factors associated with early and late death included age ≥ 65 years, heart failure, and end-stage renal disease (Table 3). 

## 4. Discussion

To the best of our knowledge, this cohort study is the first to determine the factors associated with early and late death in patients afflicted with COVID-19. Overall, the mortality rate was 12.2% among hospitalized COVID-19 patients. Of the 101 deceased, 66 (65.3%) died within two weeks of hospitalization. After adjusting for demographics and comorbidities, independent predictors of mortality included age ≥ 65 years, ICU admission, intubation, cancer, heart failure, and end-stage renal disease. When the timing of death was considered, age ≥ 65 years, ICU admission, intubation, heart failure, and end-stage renal disease were associated with a higher risk of early and late death. 

This study found that the in-hospital mortality rate was 12.2% among infectious patients, which is similar to 11.1% and 12.0% among hospitalized patients with COVID-19 in the US [6] and UK [10], respectively. Although 81% of COVID-19 cases did not require hospitalization due to mild symptoms and a low death rate [11], the mortality risk in hospitalized patients with COVID-19 was high [6]. The findings of our study suggest that hospitalized patients with COVID-19 should be treated carefully and monitored closely for their clinical condition during hospitalization.

This study found that two thirds of COVID-19 deaths occurred within two weeks of hospitalization. Moreover, in patients with early death, the median time from hospitalization to death was six days. A previous study in China demonstrated that the median time from hospital admission to acute respiratory distress syndrome in infectious patients was two days [7]. Although many patients with COVID-19 have mild symptoms [5], the clinical condition of hospitalized patients infected with SARS-CoV-2 could deteriorate rapidly [7]. The findings of our study suggest that hospitalized patients with COVID-19 should be closely monitored for disease progression during treatment.

This study showed that age ≥ 65 years was an independent predictor of early and late death in COVID-19 patients. Age-related defects in T-cell and B-cell function and the strong host innate immune response against SARS-CoV-2 may explain the high mortality in older patients with COVID-19. A previous report showed that aging was associated with the decline in T-cell and B-cell immunity, which leads to a deficiency in the control of viral replication and may cause high mortality in patients afflicted with COVID-19 [12]. In animal studies, older macaques inoculated with SARS-CoV had stronger host innate responses to viral infection than younger adults, which resulted in an increase in the differential expression of genes associated with inflammation and severe illness [13]. As older age was associated with a higher risk of mortality in patients with COVID-19, the findings of our study suggest that older adult individuals should be prioritized in the implementation of preventive measures. 

This study found that male patients with COVID-19 had a higher mortality rate than female patients. High expression of angiotensin-converting enzyme 2 (ACE2) receptor may explain the higher mortality rate in male patients with COVID-19 than female patients. Previous studies showed that the male sex was associated with an increased expression of ACE2 [14,15]. Pinto et al. found that higher ACE2 expression was associated with severe COVID-19 infection [16]. The findings of our study suggest that male patients with COVID-19 should be treated carefully and closely monitored for their clinical condition.

This study found that heart failure and end-stage renal disease were independent risk factors for early death in patients with COVID-19. Our study findings have important implications for the allocation of antiviral drugs for COVID-19 management. Currently, the US Food and Drug Administration has approved, on an emergency basis, several antiviral drugs to treat COVID-19 (e.g., sotrovimab, molnupiravir, and paxlovid), which could significantly reduce the risk of mortality in infectious patients [17,18,19]. However, due to the high demand and large cost of these new antiviral drugs, these medications may not be available for all patients with COVID-19. Since early treatment in high-risk patients with COVID-19 could reduce the risk of disease progression [17], the findings of our study suggest that patients with heart failure or end-stage renal disease should be considered priority groups for antiviral treatment.

Our report is the first to identify factors associated with early and late death in hospitalized COVID-19 patients and found that two thirds of COVID-19 deaths occurred within two weeks of hospitalization. Nonetheless, our study has two limitations. First, some important factors (e.g., pneumonia severity and treatment regimens) associated with mortality in COVID-19 patients were not available in this study. Second, the external validity of our findings may be a concern, as almost all our participants were Taiwanese. Therefore, the generalizability of our results to other non-Asian ethnic groups requires further verification. 

## 5. Conclusions

This study found that two thirds of COVID-19 deaths occurred within two weeks of hospitalization. After adjusting for demographics and comorbidities, age ≥ 65 years, heart failure, and end-stage renal disease were found to be independent predictors of early death in patients with COVID-19. Since the clinical condition of hospitalized patients with COVID-19 could deteriorate rapidly, our study suggests that hospitalized patients with COVID-19 should be treated carefully and closely monitored for their clinical condition during the treatment, particularly in older patients and in those with comorbidities.

## Figures and Tables

**Figure 1 ijerph-19-03357-f001:**
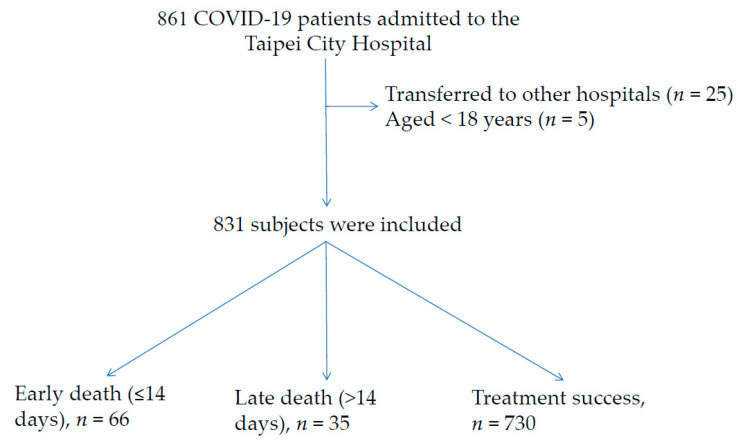
The process of enrollment in patients with COVID-19 infection.

**Table 1 ijerph-19-03357-t001:** Patients’ characteristics by treatment outcome.

Characteristics	No. (%) of Subjects *				*p* Value
	Total, *n* = 831	Early Death (≤ 14 days), *n* = 66	Late Death (> 14 days), *n* = 35	Treatment Success, *n* = 730	
Age, years					
Mean ± SD	59.3 ± 16.0	72.9 ± 11.8	72.8 ± 12.1	57.4 ± 15.6	<0.001
18–64	494 (59.45)	17 (25.76)	6 (17.14)	471 (64.52)	<0.001
≥65	337 (40.55)	49 (74.24)	29 (82.86)	259 (35.48)	
Sex					
Female	424 (51.02)	25 (37.88)	10 (28.57)	389 (53.29)	0.001
Male	407 (48.98)	41 (62.12)	25 (71.43)	341 (46.71)	
Comorbidities					
Cancer	12 (1.44)	2 (3.03)	2 (5.71)	8 (1.10)	0.043
Heart failure	10 (1.20)	4 (6.06)	2 (5.71)	4 (0.55)	<0.001
Cerebrovascular disease	17 (2.05)	1 (1.52)	2 (5.71)	14 (1.92)	0.286
Diabetes	169 (20.34)	18 (27.27)	8 (22.86)	143 (19.59)	0.309
Hypertension	225 (27.08)	34 (36.36)	7 (20.00)	194 (26.58)	0.145
End-stage of renal disease	20 (2.41)	6 (9.09)	7 (20.00)	7 (0.96)	<0.001
Follow-up days, mean (SD)	16.2 ± 11.4	6.8 ± 4.0	26.8 ± 11.1	16.5 ± 11.3	<0.001

SD, standard deviation. * Unless stated otherwise.

**Table 2 ijerph-19-03357-t002:** Univariate and multivariate analysis of factors associated with mortality among patients with COVID-19.

Variables	Number of Patients	Death During Hospitalization	Univariate	Multivariate Analysis
		*n* (%)	OR (95% CI)	AOR (95% CI)
Age, years				
18–64	494	23 (4.66)	1	1
≥65	337	78 (23.15)	6.17 (3.78–0.06) ***	6.47 (3.80–11.02) ***
Sex				
Female	424	35 (8.25)	1	1
Male	407	66 (16.22)	2.15 (1.39–3.32) ***	1.57 (0.97–2.54)
Comorbidities				
Cancer				
No	819	97 (11.84)	1	1
Yes	12	4 (33.33)	3.72 (1.10–12.59) *	3.32 (0.87–12.70)
Heart failure				
No	821	95 (11.57)	1	1
Yes	10	6 (60.00)	11.46 (3.18–41.36) ***	11.67 (2.87–47.49) ***
Cerebrovascular disease				
No	814	98 (12.04)	1	1
Yes	17	3 (17.65)	1.57 (0.44–5.55)	1.24 (0.33–4.64)
Diabetes				
No	662	75 (11.33)	1	1
Yes	169	26 (15.38)	1.42 (0.88–2.30)	0.96 (0.55–1.69)
Hypertension				
No	606	70 (11.55)	1	1
Yes	225	31 (13.78)	1.22 (0.78–1.93)	0.71 (0.42–1.20)
End-stage of renal disease				
No	811	88 (10.85)	1	1
Yes	20	13 (65.00)	15.26 (5.93–39.26) ***	18.67 (6.42–54.30) ***

* < 0.05; *** < 0.001; COVID-19, coronavirus disease 2019; AOR, adjusted odds ratio; CI, confident.

**Table 3 ijerph-19-03357-t003:** Multinomial regression analyses of risk factors for early and late death in patients with COVID-19 ^#^.

Factors	Early Death		Late Death	
	AOR (95% CI)	*p* Value	AOR (95% CI)	*p* Value
Age (years)				
18–64	1		1	
≥65	5.27 (2.88–9.65)	<0.001	10.40 (3.97–27.21)	<0.001
Heart failure	10.32 (2.28–46.65)	<0.001	15.78 (2.37–105.02)	<0.001
End-stage of renal disease	11.97 (3.53–40.55)	<0.001	41.42 (11.11–154.36)	<0.001

^#^ Reference is COVID-19 patients with successful treatment; COVID-19, coronavirus disease 2019; AOR, adjusted odds ratio; CI, confident interval.

## Data Availability

The datasets produced and analyzed during the present study are available from the corresponding author upon reasonable request.

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
