# Peer review of "Predictors for Early and Late Death in Adult Patients with COVID-19: A Cohort Study"

_ijerph, 2022, doi:10.3390/ijerph19063357_

Round 1
Reviewer 1 Report
In this cohort study, authors investigated the risk factors for early and late death in COVID-19 patients, focusing on demographics, comorbidities and severity of illness. However, they did not take into consideration the evaluation of the pneumonia severity at admission, knowing that is the main manifestation of COVID-19. As extensively described in literature, it is strongly correlated with the development of severe disease (e.g. ICU admission and intubation) and it can be assessed with clinical and radiological scores.
Minor comments:
- Section 2.2: deeper description of inclusion/exclusion criteria
- Section 2.3: did authors consider other treatments that could have an influence on the outcome (e.g. drugs)?
- Table 1 -3: significant p-values should be underlined with a bold type or an asterisk *
- Lines 175-177 should be deleted from the paragraph of limitations
- Discussion: please add more recent references to compare your results
Author Response
Reply to the Reviewer 1 Comments
1: In this cohort study, authors investigated the risk factors for early and late death in COVID-19 patients, focusing on demographics, comorbidities and severity of illness. However, they did not take into consideration the evaluation of the pneumonia severity at admission, knowing that is the main manifestation of COVID-19. As extensively described in literature, it is strongly correlated with the development of severe disease (e.g. ICU admission and intubation) and it can be assessed with clinical and radiological scores.
Reply 1: We really appreciate your comment. Since this study was a retrospective study design, some factors associated with COVID-19 mortality were not available, including the pneumonia severity at admission, treatment regimens, individuals’ body mass index. We add this information in the limitation section “First, since this study was a retrospective study design, some factors associated with COVID-19 mortality were not available, including the pneumonia severity at admission, treatment regimens, individuals’ body mass index.” Please see Page 7 (third paragraph, line 208-211).
2: Section 2.2: deeper description of inclusion/exclusion criteria
Reply 2: We really appreciate your comment. This cohort study consecutively recruited patients aged 18 years or older who had received a COVID-19 diagnosis and were admitted to TCH between May 14 and July 30, 2021. Patients with COVID-19 transferred to other hospitals or those aged < 18 years were excluded from this study. We add this information in the method “This cohort study consecutively recruited patients aged 18 years or older who had received a COVID-19 diagnosis and were admitted to TCH between May 14 and July 30, 2021. The diagnosis was confirmed using a positive real-time reverse-transcriptase polymerase chain reaction (RT-PCR) test. Patients with COVID-19 ftransferred to other hospitals or those aged < 18 years were excluded from this study.” Please see Page 2 (third paragraph, line 74-80).
3: Section 2.3: did authors consider other treatments that could have an influence on the outcome (e.g. drugs)?
Reply 3: We really appreciate your comment. Since this study was a retrospective study design, some factors associated with COVID-19 mortality were not available, including the pneumonia severity at admission, treatment regimens, individuals’ body mass index. We add this information in the limitation section “First, since this study was a retrospective study design, some factors associated with COVID-19 mortality were not available, including the pneumonia severity at admission, treatment regimens, individuals’ body mass index.” Please see Page 7 (third paragraph, line 208-211).
4: Table 1 -3: significant p-values should be underlined with a bold type or an asterisk *
Reply 4: We really appreciate your comment. The significant p-values for the factors associated with mortality in Table 2 was presented as asterisk *. Please see the table 2.
5: Lines 175-177 should be deleted from the paragraph of limitations
Reply 5: We really appreciate your comment. We delete the information in Lines 175-177.
6: Discussion: please add more recent references to compare your results
Reply 6: We really appreciate your comment. We add more references to compare our results with previous reports. Please see the revised manuscript.
Reviewer 2 Report
Thank you for giving me the opportunity to read and comment a report “Predictors for Early and Late Death in Adult Patients with COVID-19: A Cohort Study”, by Yung-Feng Yen, et al.
In the reviewed manuscript, the factors associated with early and late death in hospitalized COVID-19 patients in Taipei has been investigated.
This paper is correctly structured and with a suitable research concept. Furthermore, the study limitations are addressed.
This is a potentially interesting report but at present it is not suitable for publication.
- It would be appropriate that the categorical variables represented in terms of frequency distribution are accompanied by their corresponding confidence intervals.
- One would like to see a survival presentation/analysis for different groups of patients.
- The role of gender on clinical expression and disease outcomes in COVID-19 should be discussed
- There has been a great advance in this field in terms of mortality predictors in COVID-19 patients since the beginning of the pandemic. Among them, existing chronic conditions or comorbidities have been well-studied. Other factors including BMI, physical activity, sleep, and other earlier life lifestyle factors etc. have also been the research focus. No data were however reported here in this study about this factors. Including these data for further analysis should be encouraged in order to better elucidate their relationship and to help advance the understanding of the underlying mechanism.
- If this information is not available, it should be discussed and included as a limitation of the study. Non-inclusion of this type of variable could confound the observed association. Just something to suggest that the authors have thought about this and also in future directions. Otherwise, it seems unclear to me what the current study can say, given the somewhat already known knowledge.
Author Response
Reply to the Reviewer 2 Comments
1: It would be appropriate that the categorical variables represented in terms of frequency distribution are accompanied by their corresponding confidence intervals.
Reply 1: We really appreciate your comment. We compare the proportion of each outcome according to individuals’ demographics and comorbidities in table 1. Then we use Pearson’s χ2 test to analyze the categorical data, as appropriate. We add this information in the method “The proportion of each outcome was compared according to individuals’ demographics and comorbidities. Then, we use Pearson’s χ2 test to analyze the categorical data, as appropriate.” Please see Page 3 (first paragraph, line 98-100).
2: One would like to see a survival presentation/analysis for different groups of patients
Reply 2: We really appreciate your comment. We conducted a subgroup analysis to determine the factors associated with mortality among COVID-19 patients after stratifying the patients according to sex. After controlling for demographics, comorbidities, and severity of COVID-19, age ≥ 65 years, ICU admission, and intubation were the risk factors associated with mortality in male and female patients with COVID-19. Moreover, end-stage renal disease was a predictor for mortality in female patients with COVID-19. We add this information in the result section “Table 3 presents the subgroup analysis for the factors associated with mortality among patients with COVID-19 after stratifying the patients according to sex. After controlling for demographics, comorbidities, and severity of COVID-19, age ≥ 65 years, ICU admission, and intubation were the risk factors associated with mortality in male and female patients with COVID-19. Moreover, end-stage renal disease was a predictor for mortality in female patients with COVID-19.” Please see Page 3 (first paragraph, line 148-154).
|
Table 3. Multivariate analysis of factors associated with mortality among male and female patients with COVID-19ta |
|||||
|
Factors |
Multivariate analysis (men) |
Multivariate analysis (women) |
|||
|
AOR (95% CI) |
P value |
AOR (95% CI) |
P value |
||
|
Age (years) |
|||||
|
18-64 |
1 |
1 |
|||
|
≥65 |
11.43 (3.33-39.21) |
<.001 |
3.90 (1.87-8.13) |
<.001 |
|
|
ICU admission |
25.18 (2.19-288.84) |
0.001 |
51.00 (4.86-534.83) |
0.001 |
|
|
Intubation |
9.10 (2.78-29.78) |
<.001 |
8.98 (3.58-22.50) |
<.001 |
|
|
End-stage of renal disease |
(-) |
(-) |
|
6.11 (1.61-23.24) |
0.008 |
|
COVID-19, coronavirus disease 2019; AOR, adjusted odds ratio; CI, confident interval; ICU, intensive care unit. |
|||||
3: The role of gender on clinical expression and disease outcomes in COVID-19 should be discussed
Reply 3: We really appreciate your comment. This study found that male patients with COVID-19 had a higher mortality rate than female patients. High expression of angiotensin-converting–enzyme 2 (ACE2) receptor may explain the higher mortality rate in male patients with COVID-19 than female patients. Previous studies showed that the male sex was associated with an increased expression of ACE2. Pinto et al. found that higher ACE2 expression was associated with severe COVID-19 infection. The findings of our study suggest that male patients with COVID-19 should be treated carefully and closely monitored for their clinical condition. We add this information in the discussion section “This study found that male patients with COVID-19 had a higher mortality rate than female patients. High expression of angiotensin-converting–enzyme 2 (ACE2) receptor may explain the higher mortality rate in male patients with COVID-19 than female patients. Previous studies showed that the male sex was associated with an increased expression of ACE2 (11, 12). Pinto et al. found that higher ACE2 expression was associated with severe COVID-19 infection (13). The findings of our study suggest that male patients with COVID-19 should be treated carefully and closely monitored for their clinical condition.” Please see Page 6 (seven paragraph, line 197-203).
4: There has been a great advance in this field in terms of mortality predictors in COVID-19 patients since the beginning of the pandemic. Among them, existing chronic conditions or comorbidities have been well-studied. Other factors including BMI, physical activity, sleep, and other earlier life lifestyle factors etc. have also been the research focus. No data were however reported here in this study about this factors. Including these data for further analysis should be encouraged in order to better elucidate their relationship and to help advance the understanding of the underlying mechanism.
Reply 4: We really appreciate your comment. Since this study was a retrospective study design, some factors associated with COVID-19 mortality were not available, including the pneumonia severity at admission, treatment regimens, individuals’ body mass index. We add this information in the limitation section “First, since this study was a retrospective study design, some factors associated with COVID-19 mortality were not available, including the pneumonia severity at admission, treatment regimens, and individuals’ body mass index.” Please see Page 7 (third paragraph, line 208-211).
5: If this information is not available, it should be discussed and included as a limitation of the study. Non-inclusion of this type of variable could confound the observed association. Just something to suggest that the authors have thought about this and also in future directions. Otherwise, it seems unclear to me what the current study can say, given the somewhat already known knowledge
Reply 5: We really appreciate your comment. Since this study was a retrospective study design, some factors associated with COVID-19 mortality were not available, including the pneumonia severity at admission, treatment regimens, and individuals’ body mass index. We add this information in the limitation section “First, since this study was a retrospective study design, some factors associated with COVID-19 mortality were not available, including the pneumonia severity at admission, treatment regimens, and individuals’ body mass index.” Please see Page 7 (third paragraph, line 208-211).
Reviewer 3 Report
Interesting paper. To improve the overall quality some minor corrections are needed:
- line 40, The long-standing COVID or post-COVID-19 syndrome first gained widespread recognition among social support groups and later in the scientific and medical communities. This disease is poorly understood as it affects COVID-19 survivors at all levels of disease severity, including young adults, children, and those out of hospital. Although the precise definition of long COVID-19 may be missing, the most common symptoms reported in many studies are fatigue and wheezing that last months after acute COVID-19. please cite doi:10.1080/23744235.2021.1924397
line 42, the very high contagiousness of the virus has forced the world population to use preventive and protective measures, both for the general population and for healthcare personnel. please cite https://doi.org/10.7416/ai.2021.2439
Methods
- please add a flow diagram using the consort model
Discussion
An interesting age-specific COVID-19-associated death data from 45 countries and the results of 22 seroprevalence studies was used to investigate the consistency of infection and fatality patterns across multiple countries. The authors found that the age distribution of deaths in younger age groups (less than 65 years of age) was very consistent across different settings and demonstrate how these data could provide robust estimates of the share of the population that has been infected. They estimated that the infection fatality ratio is lowest among 5-9-year-old children, with a log-linear increase by age among individuals older than 30 years. please cite doi:10.1038/s41586-020-2918-0
I see how it has innumerable strengths as a large enrolled sample, a correct research protocol with well-made statistics. The analysis of the affected variables includes selected right influencing factors. Among other things, the ethics committee was also provided. Probably the study limitation section could be improved and implemented as the authors only hint in two sentences that the population is Taiwanese only. However apart from that it is really well done and interesting.
Author Response
Reply to the Reviewer 3 Comments
1: line 40, The long-standing COVID or post-COVID-19 syndrome first gained widespread recognition among social support groups and later in the scientific and medical communities. This disease is poorly understood as it affects COVID-19 survivors at all levels of disease severity, including young adults, children, and those out of hospital. Although the precise definition of long COVID-19 may be missing, the most common symptoms reported in many studies are fatigue and wheezing that last months after acute COVID-19. please cite doi:10.1080/23744235.2021.1924397
Reply 1: We really appreciate your comment. SARS-CoV-2 infection could cause long COVID-19 syndrome, including symptoms of fatigue and dyspnea lasting for months after the acute infection. We add this information in the introduction “SARS-CoV-2 infection could cause long COVID-19 syndrome, including symptoms of fatigue and dyspnea lasting for months after the acute infection (2).” Please see Page 1 (first paragraph, line 43-44).
2: line 42, the very high contagiousness of the virus has forced the world population to use preventive and protective measures, both for the general population and for healthcare personnel. please cite https://doi.org/10.7416/ai.2021.2439
Reply 2: We really appreciate your comment. Since SARS-CoV-2 is highly contagious, personal protective equipment (PPE) use is extensively recommended for the general population and healthcare workers to prevent the spread of the virus and the COVID-19 infection. We add this information in the introduction “Since SARS-CoV-2 is highly contagious, personal protective equipment (PPE) use is extensively recommended for the general population and healthcare workers to prevent the spread of the virus and the COVID-19 infection (3, 4).” Please see Page 1 (second paragraph, line 44-47).
3: please add a flow diagram using the consort model
Reply 3: We really appreciate your comment. We add a figure to show the process of enrollment in patients with COVID-19 infection. Please see the figure 1.
Figure 1. The process of enrollment in patients with COVID-19 infection.
4: An interesting age-specific COVID-19-associated death data from 45 countries and the results of 22 seroprevalence studies was used to investigate the consistency of infection and fatality patterns across multiple countries. The authors found that the age distribution of deaths in younger age groups (less than 65 years of age) was very consistent across different settings and demonstrate how these data could provide robust estimates of the share of the population that has been infected. They estimated that the infection fatality ratio is lowest among 5-9-year-old children, with a log-linear increase by age among individuals older than 30 years. please cite doi:10.1038/s41586-020-2918-0
Reply 4: We really appreciate your comment. O’Driscoll et al. found that increasing age was associated with higher risk of mortality in COVID-19 patients. We add this information in the discussion section “This study showed that age ≥ 65 years was an independent predictor of early and late death in COVID-19 patients. A previous report found that increasing age was associated with a higher risk of mortality in these patients (12).” Please see Page 6 (fourth paragraph, line 185-186).
5: I see how it has innumerable strengths as a large enrolled sample, a correct research protocol with well-made statistics. The analysis of the affected variables includes selected right influencing factors. Among other things, the ethics committee was also provided. Probably the study limitation section could be improved and implemented as the authors only hint in two sentences that the population is Taiwanese only. However apart from that it is really well done and interesting.
Reply 5: We really appreciate your comment. Since this study was a retrospective study design, some factors associated with COVID-19 mortality were not available, including the pneumonia severity at admission, treatment regimens, and individuals’ body mass index. We add this information in the limitation section “Our study has two limitations. First, since this study was a retrospective study design, some factors associated with COVID-19 mortality were not available, including the pneumonia severity at admission, treatment regimens, and individuals’ body mass index. Second, the external validity of our findings may be a concern as almost all our participants were Taiwanese. Therefore, the generalizability of our results to other non-Asian ethnic groups requires further verification.” Please see Page 7 (third paragraph, line 208-213).
Reviewer 4 Report
In this paper, authors investigated the predictors of death in COVID 19 patients, trying to understand the factors associated with mortality with respect to the different timings of death.
The paper is interesting and some concerns could be addressed:
- authors found that age ≥ 65 years, ICU admission, cancer, heart failure, end-stage renal disease and intubation are the predictors of death. Conversely, arterial hypertension didn't result significant statistical different among the 3 groups. This result appeared in contrast with the previous studies already published (see for example "Gallo G. Hypertension and COVID-19: Current Evidence and Perspectives. High Blood Press Cardiovasc Prev. 2022 Feb 20. doi: 10.1007/s40292-022-00506-9"). For this reason, I would suggest to try to explicate this data in the Discussion with the fact that the threshold values of blood pressure could not be coincident with the data of the real world, as well expressed here: "Di Nora C. et al Systolic blood pressure target in systemic arterial hypertension: Is lower ever better? Results from a community-based Caucasian cohort. Eur J Intern Med. 2018 Feb;48:57-63. doi: 10.1016/j.ejim.2017.08.029";
- it would be interesting to have some data on medical drugs in these patients, not only related to the previous medical treatment (if present, in patients with heart failure or chronic renal disease) but also to the COVID 19 infections, if absent, please add this in the Limitations;
- authors did not refer to another important data, that is obesity, as previuos demonstrated ("Carmona-Pírez J. Identifying multimorbidity profiles associated with COVID-19 severity in chronic patients using network analysis in the PRECOVID Study. Sci Rep. 2022 Feb 18;12(1):2831. doi: 10.1038/s41598-022-06838-9. PMID: 35181720"). I would suggest to discuss the lack of information on BMI for the patients analyzed, commenting on the fact that they did not consider the BMi because it has some limitation, as well expressed here: "Antonini-Canterin F, Obesity, Cardiac Remodeling, and Metabolic Profile: Validation of a New Simple Index beyond Body Mass Index. J Cardiovasc Echogr. 2018 Jan-Mar;28(1):18-25. doi: 10.4103/jcecho.jcecho_63_17. PMID: 29629255; PMCID: PMC5875131".
Author Response
Reply to the Reviewer 4 Comments
1: authors found that age ≥ 65 years, ICU admission, cancer, heart failure, end-stage renal disease and intubation are the predictors of death. Conversely, arterial hypertension didn't result significant statistical different among the 3 groups. This result appeared in contrast with the previous studies already published (see for example "Gallo G. Hypertension and COVID-19: Current Evidence and Perspectives. High Blood Press Cardiovasc Prev. 2022 Feb 20. doi: 10.1007/s40292-022-00506-9"). For this reason, I would suggest to try to explicate this data in the Discussion with the fact that the threshold values of blood pressure could not be coincident with the data of the real world, as well expressed here: "Di Nora C. et al Systolic blood pressure target in systemic arterial hypertension: Is lower ever better? Results from a community-based Caucasian cohort. Eur J Intern Med. 2018 Feb;48:57-63. doi: 10.1016/j.ejim.2017.08.029"
Reply 1: We really appreciate your comment. Gallo et al. reported that hypertension does not play an independent role in COVID-19 progression and severity.
2: it would be interesting to have some data on medical drugs in these patients, not only related to the previous medical treatment (if present, in patients with heart failure or chronic renal disease) but also to the COVID 19 infections, if absent, please add this in the Limitations
Reply 2: We really appreciate your comment. Since this study was a retrospective study design, some factors associated with COVID-19 mortality were not available, including the pneumonia severity at admission, treatment regimens, and individuals’ body mass index. We add this information in the limitation section “First, since this study was a retrospective study design, some factors associated with COVID-19 mortality were not available, including the pneumonia severity at admission, treatment regimens, and individuals’ body mass index.” Please see Page 7 (third paragraph, line 208-211).
3: authors did not refer to another important data, that is obesity, as previuos demonstrated ("Carmona-Pírez J. Identifying multimorbidity profiles associated with COVID-19 severity in chronic patients using network analysis in the PRECOVID Study. Sci Rep. 2022 Feb 18;12(1):2831. doi: 10.1038/s41598-022-06838-9. PMID: 35181720"). I would suggest to discuss the lack of information on BMI for the patients analyzed, commenting on the fact that they did not consider the BMi because it has some limitation, as well expressed here: "Antonini-Canterin F, Obesity, Cardiac Remodeling, and Metabolic Profile: Validation of a New Simple Index beyond Body Mass Index. J Cardiovasc Echogr. 2018 Jan-Mar;28(1):18-25. doi: 10.4103/jcecho.jcecho_63_17. PMID: 29629255; PMCID: PMC5875131"
Reply 3: We really appreciate your comment. Since this study was a retrospective study design, some factors associated with COVID-19 mortality were not available, including the pneumonia severity at admission, treatment regimens, individuals’ body mass index. We add this information in the limitation section “First, since this study was a retrospective study design, some factors associated with COVID-19 mortality were not available, including the pneumonia severity at admission, treatment regimens, individuals’ body mass index.” Please see Page 7 (third paragraph, line 208-211).
Round 2
Reviewer 1 Report
Authors improved some aspects of the paper according to my suggestions.
However, as already reported in the reviews, the absence of some factors associated with COVID-19 severity, mainly the absence of pneumonia severity and treatment regimen, is a major flaw of the study that could have influenced the reliability of the results.
Author Response
Reply to the Reviewer 1 Comments
1: Authors improved some aspects of the paper according to my suggestions.
However, as already reported in the reviews, the absence of some factors associated with COVID-19 severity, mainly the absence of pneumonia severity and treatment regimen, is a major flaw of the study that could have influenced the reliability of the results.
Reply 1: We really appreciate your comment. Although our report is the first to identify factors associated with early and late death in hospitalized COVID-19 patient, the information regarding the patients’ pneumonia severity and treatment regimens were not available in this study. We add this information in the limitation section “Our report is the first to identify factors associated with early and late death in hospitalized COVID-19 patients and found that two-thirds of COVID-19 deaths occurred within two weeks of hospitalization. Nonetheless, our study has two limitations. First, some important factors (e.g., pneumonia severity and treatment regimens) associated with mortality in COVID-19 patients were not available in this study.” Please see Page 7 (third paragraph, line 224-228).
Reviewer 2 Report
The manuscript has been improved and is now suitable for publication
Author Response
Reply to the Reviewer 2 Comments
1: The manuscript has been improved and is now suitable for publication.
Reply 1: We really appreciate your comment.
Reviewer 4 Report
The paper is suitable for publication in the present form.
Author Response
Reply to the Reviewer 4 Comments
1: The paper is suitable for publication in the present form.
Reply 1: We really appreciate your comment.